# Research on Three-Dimensional Morphological Characteristics Evaluation Method and Processing Quality of Coarse Aggregate

Weixiong Li [1,2], Duanyi Wang [1], Bo Chen [1,2,*], Kaihui Hua [3,*], Wenzheng Su [3], Chunlong Xiong [1] and Xiaoning Zhang [1,2]

1 School of Civil Engineering and Transportation, South China University of Technology, Guangzhou 510006, China; 201810101689@mail.scut.edu.cn (W.L.); tcdywang@scut.edu.cn (D.W.); cthgclx@mail.scut.edu.cn (C.X.); ctxnzh@scut.edu.cn (X.Z.)
2 Guangzhou Xiaoning Roadway Engineering Technology Research Institute Co., Ltd., Wushan Road, Guangzhou 510641, China
3 School of Environment and Civil Engineering, Dongguan University of Technology, Dongguan 523808, China; 2112008002@dgut.edu.cn
* Correspondence: chenb@scut.edu.cn (B.C.); huakh@dgut.edu.cn (K.H.)

**Abstract:** The size, shape, gradation and appearance of aggregate have a significant impact on the road performance of asphalt mixtures, which is directly related to the deformation characteristics and fatigue resistance of asphalt mixtures. In order to be able to design a long-life asphalt pavement, the aggregate must have reasonable morphology and morphological characteristics. In order to quantitatively analyze the shape characteristics of the road coarse aggregate, a high-precision three-dimensional scanner is used to obtain the characteristic lattice of the aggregate surface, and the solid model of the coarse aggregate particles is established. The two-dimensional and three-dimensional morphological indicators of the aggregate are analyzed and discussed. Meanwhile, aggregates processed by typical quarries in Guangdong Province are collected, and the influence of different processing techniques on the morphology of aggregates are analyzed. The results show that the difference between the perimeter and projected area of the aggregate contour under different viewing angles is relatively large, which is closely related to the flatness index of the aggregate. It can better characterize the three-dimensional shape of the aggregate. The closer the aggregate is to the cubic state, the greater the sphericity value; the ellipsoid index calculated based on the three-dimensional circumscribed ellipsoid can better characterize the angularity of the aggregate. The worse the angularity of the aggregate, the larger the corresponding ellipsoid value. The sphericity of the aggregate processed by counter-breaking is lower, and the angularity is better. The sphericity of the aggregate processed by the shaping process is the best, but the angularity is lower. According to actual needs, different processing techniques can be combined and blended to obtain aggregate finished products with a more balanced grain shape and angularity. The richer the angularity of the coarse aggregate, the better the high-temperature stability and fatigue resistance of the asphalt mixture. However, the stability of performance indicators will become worse. In practical engineering applications, it is recommended to further combine the screening efficiency of the hot material screen of the mixing plant with the stability of the hot material gradation and the uniformity of construction quality to select a suitable aggregate processing technology.

**Keywords:** coarse aggregate; three-dimensional scanning; two-dimensional profile; sphericity; ellipticity; processing technology

## 1. Introduction

Natural aggregate is the most widely used material in highway construction, and its mass proportion in asphalt pavement can usually reach more than 90%. Therefore, considering the proportion of material quality, the quality of an asphalt pavement is largely determined by the quality of the aggregates. Engineers and technicians generally believe

that the performance of an asphalt pavement mainly depends on the quality of the asphalt binder, which only accounts for 5% to 10% of the weight of asphalt concrete. In order to change the traditional and limited view that "asphalt performance determines pavement performance", it is necessary to systematically study aggregates and analyze the influence of aggregate properties on the performance of the mixture [1].

For a long time, construction management personnel have also had prejudices in the understanding of aggregates, that is, they have strict requirements on the original properties of aggregates, such as the type of aggregate, crushing value, abrasion value, polishing value, density and water absorption. However, they do not pay enough attention to the processing characteristics, such as the content of needle and flake particles of stone, the proportion of broken surface of broken gravel, mud content, angularity, grading composition, and the roughness and angularity of machine-made sand [2]. Previous studies have shown that the size, shape, gradation and morphology of aggregate have a significant impact on the road performance of asphalt mixture, which is directly related to the deformation characteristics and fatigue resistance of asphalt mixture [3]. In order to be able to design a long-life asphalt pavement, the aggregate must have reasonable morphology and morphological characteristics. Therefore, how to control aggregate quality from the source, evaluate the performance of the asphalt mixture and predict its road performance is an urgent problem to be solved [4].

The properties of coarse and fine aggregates used in hot-mix asphalt mixtures and cement concrete mixtures have an important impact on the performance of a pavement. Among them, the shape, texture and corners of the aggregate properties have a huge impact on the performance of the aggregate [5]. In recent years, several methods have been developed to measure the shape, texture and corners of aggregates. According to the research on aggregate testing methods, the methods of describing aggregate shape characteristics can be roughly summarized into two categories, namely, the indirect method and direct method [6].

The indirect method refers to the use of test methods to combine the characteristics of the particles, that is, to measure the overall macroscopic properties of aggregates that are stacked or formed in a certain way, such as the loose porosity of the aggregates, the internal friction angle and so on [7]. The measured overall properties are taken as the characteristic values of the particles [8]. At present, the representative indirect test methods for the morphological characteristics of aggregates include: (1) the uncompacted porosity method of coarse/fine aggregate (AASHTO TP56/T304), (2) the compacted aggregate resistance test method (CAR), (3) the Florida bearing ratio test method, (4) the direct shear test method (AASHTO T236/ASTM D3080), and (5) the gauge method (ASTM D4791) [9].

Prowell and Weingart studied the test accuracy of ASTM D4791 and found that the coefficient of variation for a single operation could reach 26.1%, and the coefficient of variation between laboratories could reach 35.3% [10]. The angularity of coarse aggregates by manually counting the fracture surfaces is evaluated in the ASTM D5821 [11]. The indirect method is time-consuming and laborious, and the accuracy of the test depends on the proficiency and experience of the operator. In addition, AASHTO TP56 and ASTM D3398 cannot separate the shape, corners and texture characteristics of aggregates [12].

The direct method refers to the use of test methods to accurately determine the specific shape of each aggregate particle, such as the use of a caliper to determine the needle-like content of the aggregate, that is, to quantitatively describe the characteristics of the particles (shape, corners and texture) [13]. At present, the representative direct test methods of aggregate morphology characteristics are: (1) the coarse aggregate broken particle percentage test method (ASTM D5821), (2) the coarse aggregate needle flake content test method (ASTM D4791) and (3) the multi-magnification rate analysis (MRA) [14].

Masad E. et al. developed the aggregate image analysis system AIMS. Based on the change in the radius of the image contour line, they proposed the form index and radius angularity index to characterize the shape and angularity of aggregates [15]. Masad E. developed the second-generation AIMS II, which eliminated the interference of ambient

light by setting a closed dark box, and used a stable LED light source to control the light intensity during the measurement process to achieve uniform exposure and scanning effects [16]. Similar to China's T0311 method (using a vernier caliper to measure particles with the ratio of maximum length to minimum thickness of coarse aggregate greater than 3 times), other standard methods commonly used in the world, such as the British BS EN 933-4 method, mainly use a particle slide gauge to measure the shape index of aggregate, expressed by the ratio of the mass of all particles greater than 3:1 to the total mass of each grade of aggregate. The flakiness index of aggregate is measured by British BS EN 933-3 method. Each grade of aggregate is screened by a reinforcement screen. The particles passing through bar sieves are expressed by the ratio of sheet mass to total mass. These methods can characterize the needle and flake particle content of aggregate and are widely used in engineering. However, there are also some limitations: first, the whole process must be completed manually, which inevitably brings subjective errors. In addition, due to the complexity of the three-dimensional shape of aggregate, it is difficult to find the accurate position of the length, width and thickness of aggregate. In contrast, the three-dimensional analysis method of aggregate can more comprehensively reflect the spatial shape of aggregate. Through the operation of the program, the maximum length, width and thickness of aggregate can also be accurately determined, and then the three-dimensional shape can be evaluated by indicators from multiple angles.CT tomography technology is currently the mainstream platform for obtaining the mesoscale of the internal structure of the mixture. A series of continuous tomographic two-dimensional slice images output by the scan are the basis for the two-dimensional feature extraction and three-dimensional information reconstruction of the aggregate [17].

Aggregate morphology has three scales: shape, angularity and texture. With the development of image and computer technology, the evaluation of aggregate morphology has basically realized the change from time-consuming, labor-intensive and subjective manual testing methods to automated objective evaluation methods based on computer technology [18]. Meanwhile, Wang et al. found that the rotational speed of VSI is affected by the input power of the machine on the particle size. In the crushing process, the higher the speed, the more cracks on the aggregate. In addition, the correlation between the particle shape characteristics affected by mineral composition and its collision behavior is discussed according to the test results [19]. Regression analysis shows that the percentage of layered silicate minerals in rocks is positively correlated with water content and total porosity. In mafic and ultramafic rock samples, the relationship between secondary layered silicate minerals and their physical and mechanical properties shows that the total amount of secondary layered silicate minerals has a negative impact on their physical and mechanical properties. On the other hand, the proportion of layered silicate minerals in volcanic rocks is low, so its engineering properties cannot be determined [20]. The ratio of secondary minerals to primary minerals (SEC/PR) of the studied ultramafic rocks has a good correlation with their physical, physicochemical and mechanical properties, indicating that alteration has a negative impact on the engineering properties of ultramafic rocks [21].

In the early stages of morphological evaluation, the two-dimensional images of aggregates were mainly used for their morphological evaluation. With the development of 3D modeling technology, the three-dimensional morphological evaluation of aggregates has become a research hotspot. Industrial CT and 3D laser scanning are the current mainstream 3D modeling methods for aggregates. In this study, the two modeling methods are used to model the coarse aggregate samples collected from nine stockyards. On this basis, the three-dimensional morphological evaluation index is constructed by using the spatial geometric characteristics of the aggregate geometric model. Finally, the relationship between the aggregate particle size, aggregate lithology and aggregate geometric characteristics is systematically studied.

## 2. Method and Equipment

### 2.1. Three-Dimensional Scanning Test of Natural Aggregate

In this study, AutoScan DS-EX 3D scanner (SHINING 3D, AutoScan DS-EX, Dongguan, China) is used, with a camera resolution of 1.3 million pixels, a scanning accuracy of ≤0.015 mm, a scanning range of 100 mm × 100 mm × 75 mm, a scanning time of 120 s and a temperature range of 10–30 °C. The size of the scanner is 260 mm × 270 mm × 420 mm, the weight is 5 kg, the data output format is STL and OBJ, the interface is USB3.0, and the power supply is DC24V. This equipment can perform splice scanning to obtain high-precision complete three-dimensional morphological data of different coarse aggregates, as shown in Figure 1.

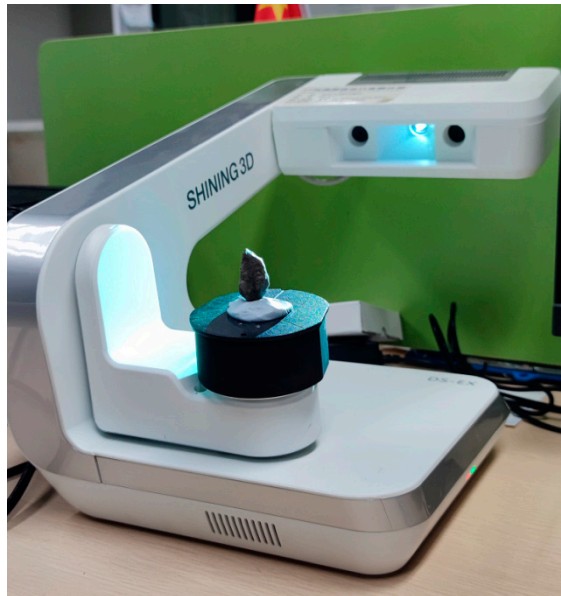

**Figure 1.** Aggregate HD Scan.

Materialise magics software (Ver 20.0.3.11, 2016, Materialise; Leuven, Belgium) is used to analyze the surface area, volume, sphericity and flatness of selected aggregates. Rhinoceros software (Ver 7.0, 2020, Robert McNeel & Assoc; Seattle, WA, USA) is used to analyze the perimeter, area, roundness and morphological factors of the selected aggregate projection.

### 2.2. Aggregate Scanning

#### 2.2.1. Aggregate Pretreatment

The working principle of the 3D scanner is to project specific light rays to the surface of the object to be measured through the scanning device, and the light is reflected by the camera of the scanning device to reproduce the 3D data of the object being measured in the software through the special algorithm of the scanning software. Therefore, the scanning device receiving the reflected light of the measured object is a necessary factor for obtaining three-dimensional data, and dark colors and reflective spots cannot be scanned during the scanning process.

Part of the surface of the crushed stone after treatment is reflective, and some components such as quartz stone appear black and the color is too dark. It is unfavorable for completing the three-dimensional scanning in the two situations. In the experiment, a thin layer of zirconia powder with a particle size of 50 nm is applied on the surface of the gravel to solve the problem of light reflection. For the spots on the gravel that are black and the powder is too thin to be easily scanned, contrast agent is applied to the aggregate to enhance the scanning effect, so as to achieve the weakening of the dark or reflective parts of the gravel and achieve the scanning conditions.

### 2.2.2. Image Mosaic

Fix the gravel on the scanning table, scan the gravel with a three-dimensional scanner to obtain the appearance data. When scanning, the environment where the gravel is located (such as a scanning table, etc.) will inevitably be scanned. The data obtained have a lot of noise data. Data processing is required to delete the noise. After removing the noise data, image stitching is performed. Because the contact point between the rubble and the scanning platform cannot be scanned, the experiment first scans the upper half of the rubble, and then reverses it to scan the lower half of the rubble. Therefore, the experiment must use the stitching scanning method, and this is carried out after the stitching is completed. The data are merged to form a complete three-dimensional image. Due to the noise in the scanning process, after the image is completed, the computer can automatically perform basic calculations based on the surrounding feature points to fill in the vacancies and form a relatively perfect gravel shape (see Figure 2).

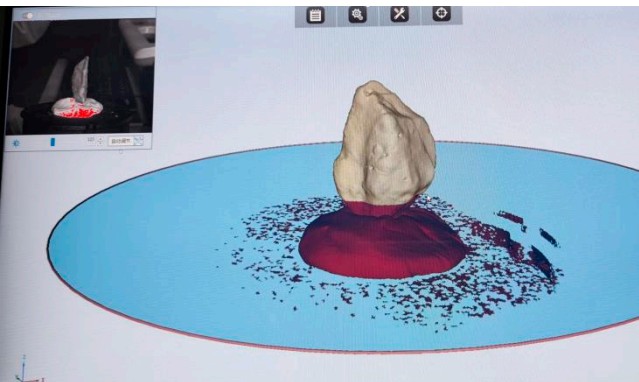

**Figure 2.** Morphological integration.

### 2.2.3. Image Restoration and 3D Reconstruction

The scanned data are mainly point cloud data. When the triangle patch is automatically generated, the point cloud data become model data, because the point cloud scan does not only acquire one layer but a multi-layer overlay form, which will have a certain degree of coincidence. This experiment uses Max software (V20.3, Autodesk company, SAN Rafael, CA, USA) to repair it. After completing the hole filling and repairing, the model is basically completed, as shown in Figure 3. Finally, it is converted to STL format or other graphic formats for export.

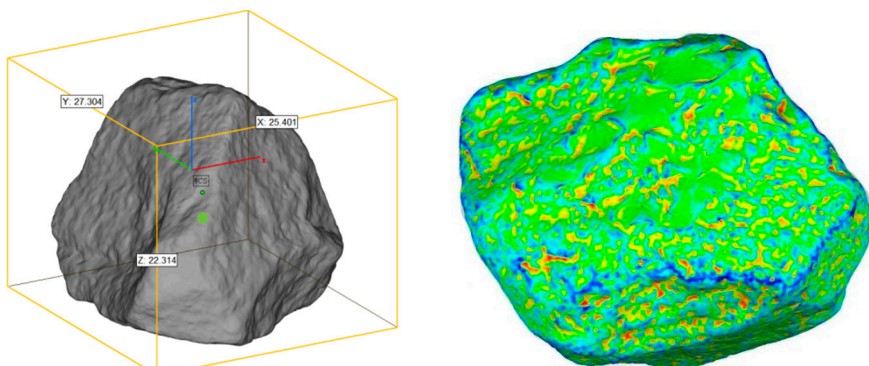

**Figure 3.** Three-dimensional reconstruction of gravel model.

## 3. Results and Discussion

### 3.1. Two-Dimensional Profile Analysis and Characterization of Aggregate

#### 3.1.1. Two-Dimensional Morphological Characterization of Aggregate

The Rhinoceros software is used to characterize the two-dimensional morphology of the crushed stone, and the three-dimensional projection map of the crushed stone is derived, and then data collection and morphological characterization are performed on the lateral projection map of the crushed stone. Fifteen aggregates are randomly selected as a sample, and the results of the perimeter and area of the projected two-dimensional images of the aggregates at different viewing angles (*x*-axis, *y*-axis and *z*-axis, as shown in Figure 4) are counted, respectively, as shown in Table 1. The projections of gravel in different directions have great differences in the measured perimeter and area. Among them, the particle size of the small gravel is 4.75~9.5 mm, the circumference of the projection surface is in the range of 30~50 mm and the projection circumference range in different directions (the difference between the maximum and minimum viewing angle) can reach more than 20 mm. The particle size of the bigger gravel is 19~26.5 mm, the perimeter of the projection surface is in the range of 50~95 mm and the range of the projection perimeter in different directions can reach more than 40 mm. From the area statistics results, the projected area of small gravel is in the range of 35~115 mm$^2$, and the maximum range of area is 77 mm$^2$. The projected area of large gravel is in the range of 150~535 mm$^2$, and the maximum range of area is 339 mm$^2$. It can be seen that even if the same gravel is viewed from different perspectives, the outline perimeter and projected area are quite different, and it is difficult for a single aggregate projected two-dimensional geometric index to characterize the morphological characteristics of the aggregate.

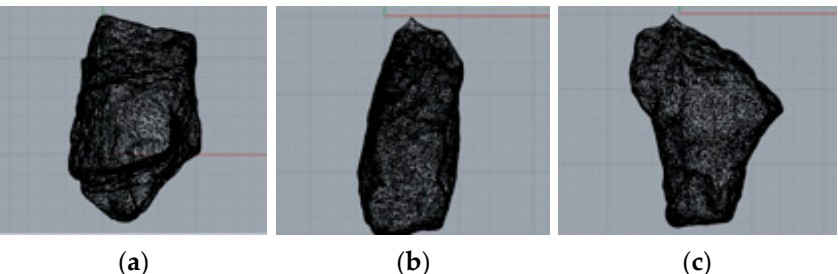

(**a**)        (**b**)        (**c**)

**Figure 4.** Aggregate projection from different angles. (**a**) *x*-axis. (**b**) *y*-axis. (**c**) *z*-axis.

**Table 1.** Three-dimensional projection test results.

| Type | Number | Perimeter/mm | | | | Area/mm$^2$ | | | |
|------|--------|--------|--------|--------|--------|--------|--------|--------|--------|
| | | *x*-axis | *y*-axis | *z*-axis | Range | *x*-axis | *y*-axis | *z*-axis | Range |
| Small gravel | 1 | 39.6 | 51.0 | 29.7 | 21.3 | 91.7 | 75.7 | 56.9 | 34.8 |
| | 2 | 40.7 | 34.3 | 58.3 | 24.1 | 115.8 | 74.0 | 57.1 | 58.8 |
| | 3 | 34.5 | 34.1 | 31.6 | 3.0 | 85.4 | 79.2 | 72.6 | 12.8 |
| | 4 | 34.3 | 34.2 | 32.7 | 1.6 | 84.6 | 86.5 | 76.2 | 10.2 |
| | 5 | 37.7 | 34.9 | 25.8 | 11.9 | 99.3 | 74.3 | 44.0 | 55.3 |
| | 6 | 42.0 | 37.0 | 26.8 | 15.2 | 113.9 | 70.2 | 36.8 | 77.1 |
| | 7 | 33.3 | 36.1 | 31.5 | 4.6 | 76.1 | 97.3 | 71.1 | 26.2 |
| | 8 | 36.3 | 33.3 | 29.4 | 6.9 | 98.6 | 79.2 | 60.3 | 38.3 |

**Table 1.** *Cont.*

| Type | Number | Perimeter/mm | | | | Area/mm² | | | |
|------|--------|--------|--------|--------|--------|--------|--------|--------|--------|
| | | *x*-axis | *y*-axis | *z*-axis | Range | *x*-axis | *y*-axis | *z*-axis | Range |
| | 1 | 81.4 | 75.9 | 86.6 | 10.7 | 447.2 | 388.2 | 534.5 | 146.3 |
| | 2 | 84.6 | 80.8 | 78.2 | 6.4 | 514.3 | 447.2 | 433.1 | 81.1 |
| | 3 | 77.8 | 68.3 | 65.6 | 12.2 | 443.3 | 328.4 | 307.4 | 135.9 |
| Larger gravel | 4 | 86.9 | 86.9 | 65.8 | 21.1 | 513.2 | 498.2 | 317.2 | 196.0 |
| | 5 | 70.7 | 75.7 | 65.6 | 10.1 | 327.6 | 414.5 | 309.1 | 105.4 |
| | 6 | 75.4 | 81.3 | 73.0 | 8.3 | 368.3 | 450.1 | 361.5 | 88.6 |
| | 7 | 93.1 | 87.3 | 50.1 | 43.1 | 489.0 | 335.2 | 149.9 | 339.1 |

### 3.1.2. Correlation Analysis of Two-Dimensional Outline Variability and Aggregate Shape

The variation in the projection shape of the aggregate from different viewing angles is mainly related to the irregularity of the aggregate. During the test operation, it is found that the results of different operators have great variability, up to more than 25%. According to the collected aggregate model, the flatness rate index of the crushed rock is calculated by extracting the long and short axis parameters of the crushed rock. The mathematical expression is

$$D = \frac{b}{a} \tag{1}$$

where: $D$ represents the flatness (slenderness ratio) of the gravel, $a$ is the length of the longest axis of the gravel, $b$ is the length of the shortest axis of the gravel and the value range of $D$ is 0–1.

According to the test regulations (JTG E42-2005), when $D$ is less than 0.333, it can be regarded as needle flake particles. The range can be used to evaluate the degree of dispersion of the two-dimensional outline index of the aggregate with the same sample number and the same size ratio, but for aggregates of different size ratios, the magnitude of the perimeter and area indicators will change due to the difference in scale. Therefore, the coefficient of variation is used to characterize the degree of dispersion of the test results of aggregate samples of different size ratios in the study. In order to study the relationship between the variability of the two-dimensional outline index of the aggregate under different viewing angles and the shape of the aggregate, the flatness rate index of the small gravel and the large gravel are calculated, respectively, and the trend charts of the variation coefficient of the perimeter and area of each aggregate and flattening rate index in different viewing angles are plotted, as shown in Figures 5 and 6.

It can be clearly seen that the coefficient of variation of samples from different viewing angles for the perimeter and area indicators of the two-dimensional outline has a good linear relationship with the flatness rate index of the aggregate. With the decrease in the flatness rate index of aggregates, the aggregate outline indexes of different viewing angles also change significantly, and the coefficient of variation of the perimeter and the area have a significant increasing trend. Combined with the morphological analysis of the aggregate, when the aggregate particles are shorter and smaller, that is, the more square the aggregate, the closer the projection shapes in all directions are. The flatter or more slender the aggregate, the greater the difference in the projection outline of the aggregate under the three-dimensional viewing angle. Therefore, the representativeness of the two-dimensional outline index of the aggregate mainly depends on the square shape of the aggregate particles. According to the 95% guarantee rate standard in engineering application, when the flattening rate of the aggregate is about 0.7 or more, the variation coefficient of the two-dimensional outline index can be controlled within 5%. For needle flake aggregate particles, the variation coefficient of the two-dimensional outline index of

the aggregate is 30–50%. At this time, the variation in the projected two-dimensional outline index is too large, and it is difficult to fully represent the morphological characteristics of the aggregate. It is suggested that for aggregates with a flattening rate of more than 0.7, conventional two-dimensional indicators can be directly used for characterization and application. In addition, when there are too many flat particles in aggregate, it is easy to cause quality problems such as production gradation variation and construction segregation of the asphalt mixture.

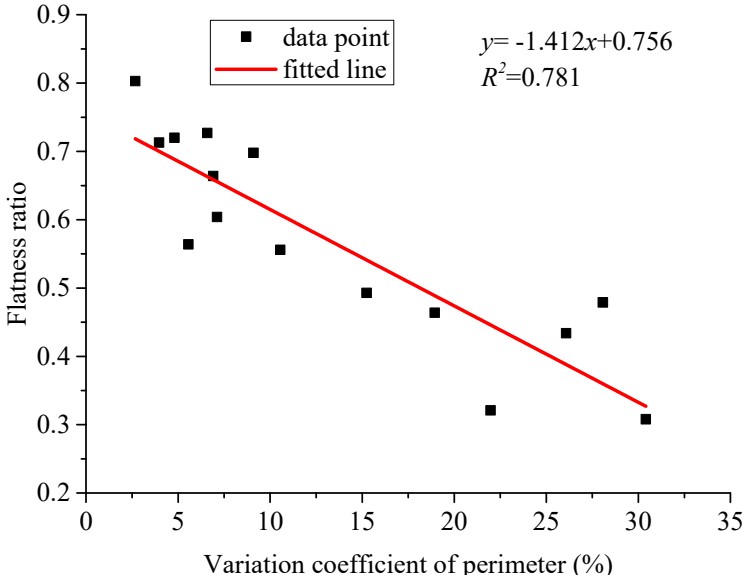

**Figure 5.** The correlation between the variation coefficient of perimeter and flat rate.

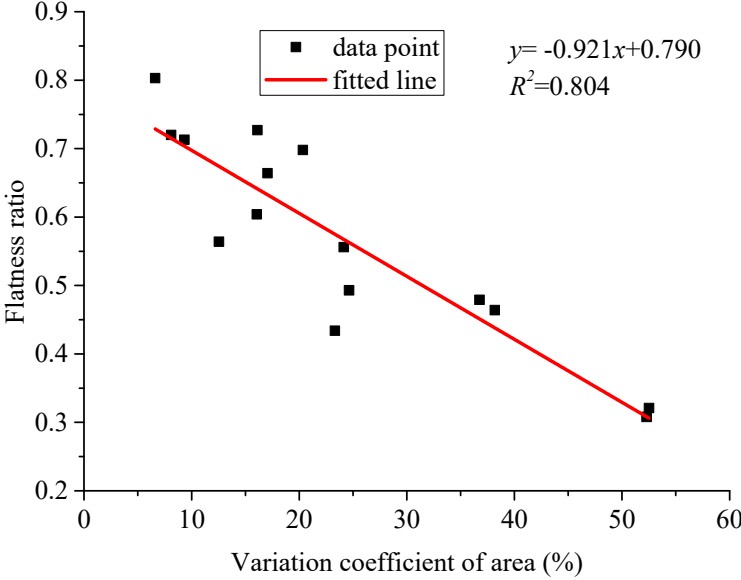

**Figure 6.** The correlation between the coefficient of variation of area and flat rate.

### 3.2. Characterization of Aggregate Particle Size

We collected the basic data of the crushed stone through Materialise Magics, exported the three-dimensional map of the crushed stone through the software (see Figure 7), collected the surface area, volume and other data of the crushed stone in the three-dimensional space and then were able to characterize the sphericity of the crushed stone. Sphericity is mainly used to evaluate the degree of crushed stone biased towards balls. In engineering applications, after the stone processing line is over-grinded and reshaped. The aggregate

particles tend to be cubic and the sphericity value increases. For the calculation of sphericity, refer to the calculation method of aggregate sphericity of AIMS II [22], and the calculation formula is as follows

$$SP = \sqrt[3]{\frac{dsdI}{dL}} \qquad (2)$$

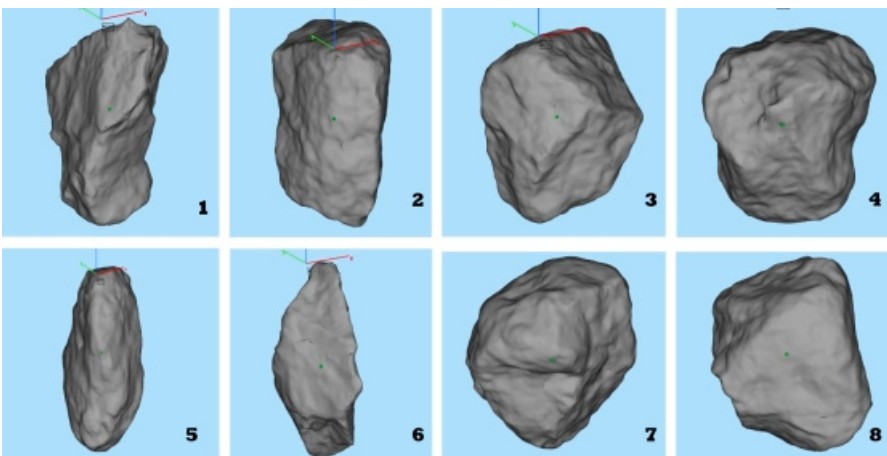

**Figure 7.** Scanning images of 8 coarse aggregate samples.

Within this formula, ds is the length of the minor axis of the aggregate in mm, dI is the length of the minor axis of the aggregate in mm and dL is the length of the major axis of a single particle in mm.

The maximum value of the sphericity index is 1. At this time, the three-axis length of the aggregate is equal, and the shape of the aggregate is similar to a cube or close to a sphere.

It can be seen from Figure 7 and Table 2 that the calculated aggregate particle sphericity index is in good agreement with the aggregate shape. At the same time, the index of flattening rate is calculated. With the increase in flattening rate, the sphericity of aggregate also shows a consistent increasing law. Both indicators can characterize the three-dimensional shape of aggregate on a certain level, but the characterization of sphericity index is more comprehensive. The reason is that the flattening rate index is mainly related to the long axis and short axis, while the calculation of sphericity index involves the length of aggregate axis in three directions, and the characterization range is closer to the three-dimensional particle shape of aggregate. Therefore, the sphericity index can better characterize the overall particle cube shape of aggregate. Combined with test conditions and engineering experience, under the conditions of this study, when the sphericity of aggregate is greater than 0.90, it is close to a cube; when $0.8 \leq$ sphericity $< 0.9$, it presents as a subcube shape; when $0.7 \leq$ sphericity $< 0.8$, the aggregate is slender; when the sphericity is less than 0.70, the flattening rate is low and reaches the degree of a needle flake. Although the sphericity index is suitable for evaluating the poor particle type of aggregate, it cannot distinguish the roughness or angular characteristics of aggregate particles. Therefore, a single sphericity index has difficulty reflecting the quality of aggregate, and the angular index of aggregate needs to be further established.

**Table 2.** Sphericity statistics of samples.

| Number | 1 | 2 | 3 | 4 | 5 | 6 | 7 | 8 |
|---|---|---|---|---|---|---|---|---|
| Sphericity | 0.725 | 0.766 | 0.901 | 0.983 | 0.678 | 0.633 | 0.881 | 0.856 |
| Flattening | 0.434 | 0.479 | 0.72 | 0.803 | 0.464 | 0.321 | 0.684 | 0.646 |
| Describe | Slender | Slender | Stereoscopic | Stereoscopic | Needle flake | Needle flake | Sub stereo-scopic | Sub stereo-scopic |

### 3.3. Evaluation of Angularity of Aggregate

The angularity of the aggregate helps to improve the stability of the compaction between the aggregates. Reference [23] proposed to use the ratio of the perimeter of the two-dimensional aggregate contour to the equivalent ellipse contour to define the angularity. Reference [24] proposed a three-dimensional angularity index based on the three-dimensional shape of the aggregate, which more comprehensively characterizes the three-dimensional angular shape of the aggregate (see Figure 8), and the index is unique. This is calculated as follows

$$E = \frac{V_1}{V_2} \tag{3}$$

where, $E$ is the three-dimensional angularity index; $V_1$ is the three-dimensional volume of the aggregate, mm$^3$; $V_2$ is the minimum ellipsoid volume circumscribed by the aggregate, mm$^3$.

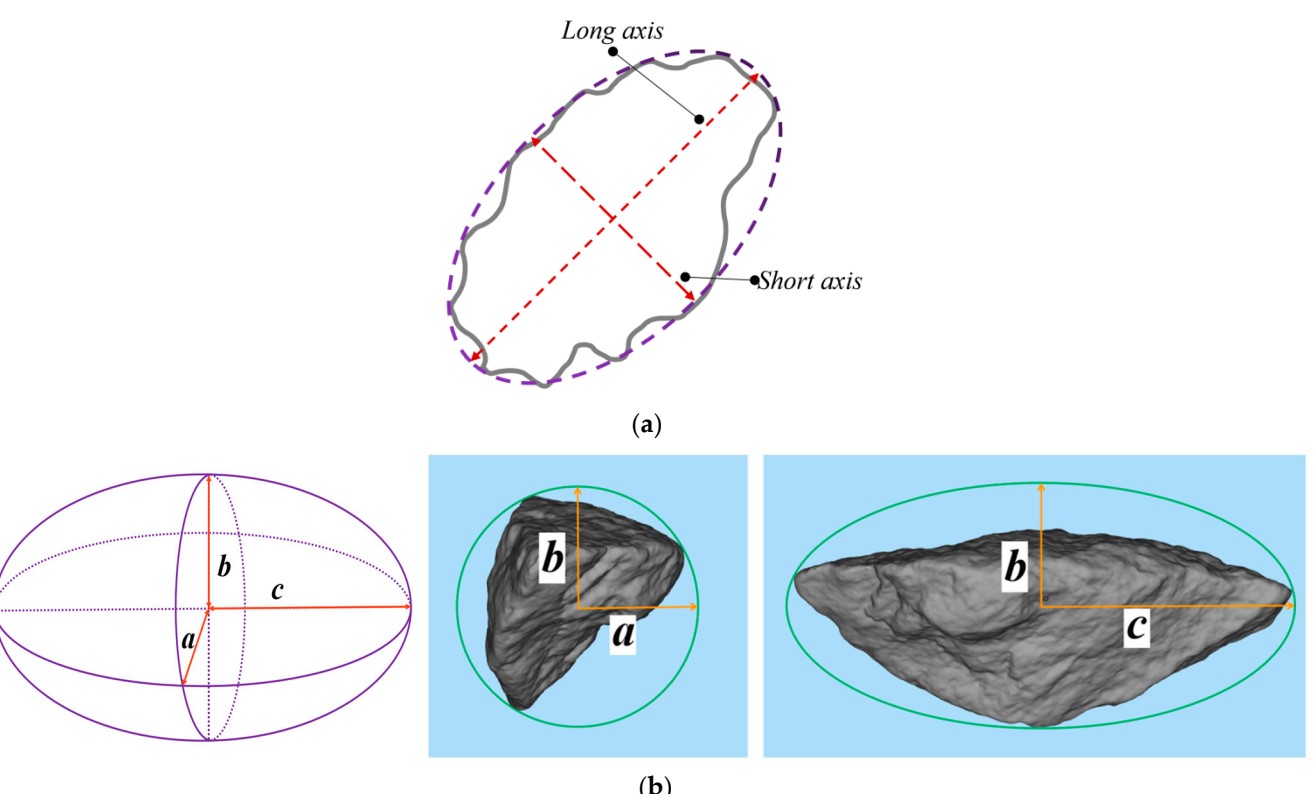

**Figure 8.** Angularity of aggregate. (**a**) Two-dimensional contour circumscribed ellipse. (**b**) Three-dimensional volume circumscribed ellipsoid.

The judgment standard is proposed in reference [25], and the angularity index of the sample is summarized in Table 3. Combining the morphology of aggregate samples and the discrimination criteria, there is a greater risk of fracture and damage for sharp and angular particles with an ellipsoid value less than 0.71. For angular particles with an ellipsoid value in the range from 0.71 to 0.79, and an ellipsoid with sub-angular particles in the range

from 0.79 to 0.88, the aggregate particles are rich in edges and corners without being too sharp, which can effectively improve the inter-aggregate embedding capacity of the asphalt mixture. When the ellipsoid value is greater than 0.88, the aggregate appears as subcircular or sub-elliptical, and the edges and corners are basically rounded, which is not conducive to the embedding effect of the asphalt mixture.

**Table 3.** Angularity statistics of samples.

| Number | 1 | 2 | 3 | 4 | 5 | 6 | 7 | 8 |
|---|---|---|---|---|---|---|---|---|
| Ellipsoid | 0.686 | 0.703 | 0.736 | 0.882 | 0.891 | 0.664 | 0.868 | 0.754 |
| Type | Sharp corners | Sharp corners | Angular | Subcircular | Subcircular | Sharp corners | Sub-angular | Angular |

*3.4. The Influence of Crushing Process on the Morphology of Coarse Aggregate*

The instability of aggregate processing quality is currently a major problem in domestic quarries. With the standardized management of high-grade highway construction in recent years, higher requirements have been placed on the grain size of aggregate processing, especially the wear layer of asphalt pavement. It is expected that the aggregate particles produced are square and rich in edges and corners to improve the performance of the road surface. In order to analyze the impact of different crushing processes on the size of coarse aggregates, the aggregate conditions processed by Furong Quarry in Guangdong Heyuan were investigated, and the aggregates of 10~15 mm specifications of three processing processes were collected, respectively: ① Process 1: counterattack crushing is the main method, that is, jaw crushing (JW1060, South Road Machinery Co., Ltd., Quanzhou, China), cone crushing (VC1500, South Road Machinery Co., Ltd., Quanzhou, China) and counterattack crushing (IH1316, South Road Machinery Co., Ltd., Quanzhou, China); ② process two: plastic crushing involves mainly jaw crushing (JW1060, South Road Machinery Co., Ltd., Quanzhou, China), cone crushing (VC1500, South Road Machinery Co., Ltd., Quanzhou, China) and shaping (S3-1030S3, South Road Machinery Co., Ltd., Quanzhou, China); ③ process three: this involves 50% counterattack crushing and 50% shaping process, that is, right semi-blended two processing aggregates.

In total, 100 aggregates under different processes were randomly selected, and the sphericity and ellipsoid index of the coarse aggregate under the three processing processes were calculated. The aggregate proportion is the ordinate, and the distribution diagram is drawn, see Figures 9–11. It can be clearly seen that the sphericity and ellipsoid values of the coarse aggregate under different processing techniques have obvious peaks, showing a kurtosis distribution. However, the kurtosis of the aggregate test results under different processes is different, and there is a partial peak state. Gaussian distribution fitting is performed on the histogram, and the statistical results are shown in Table 4.

**Table 4.** Aggregate form index statistics.

| Craft | Sphericity | | | Ellipticity | | |
|---|---|---|---|---|---|---|
| | Mean Value | Standard Deviation | $R^2$ | Mean Value | Standard Deviation | $R^2$ |
| Craft 1 | 0.734 | 0.072 | 0.868 | 0.731 | 0.071 | 0.952 |
| Craft 2 | 0.826 | 0.057 | 0.926 | 0.821 | 0.064 | 0.927 |
| Craft 3 | 0.785 | 0.059 | 0.933 | 0.777 | 0.081 | 0.975 |

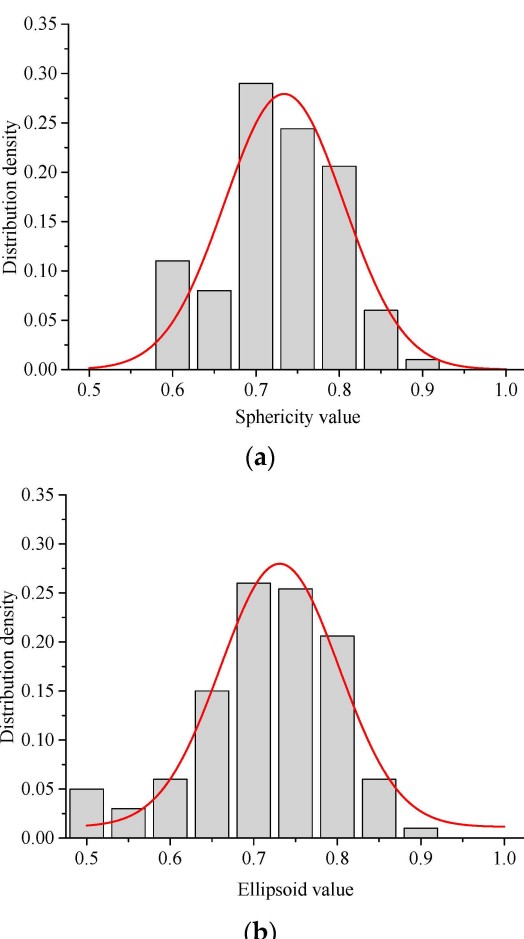

**Figure 9.** Aggregate morphology distribution of process one. (**a**) Sphericity. (**b**) Ellipticity.

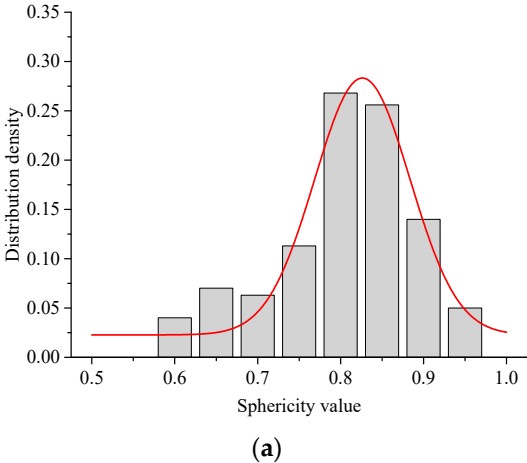

**Figure 10.** *Cont*.

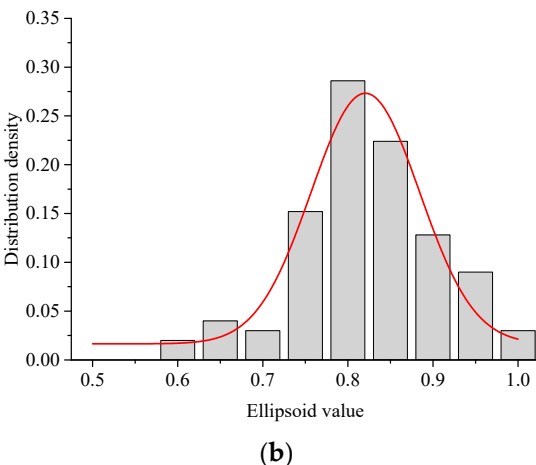

**(b)**

**Figure 10.** Aggregate morphology distribution of process two. (**a**) Sphericity. (**b**) Ellipticity.

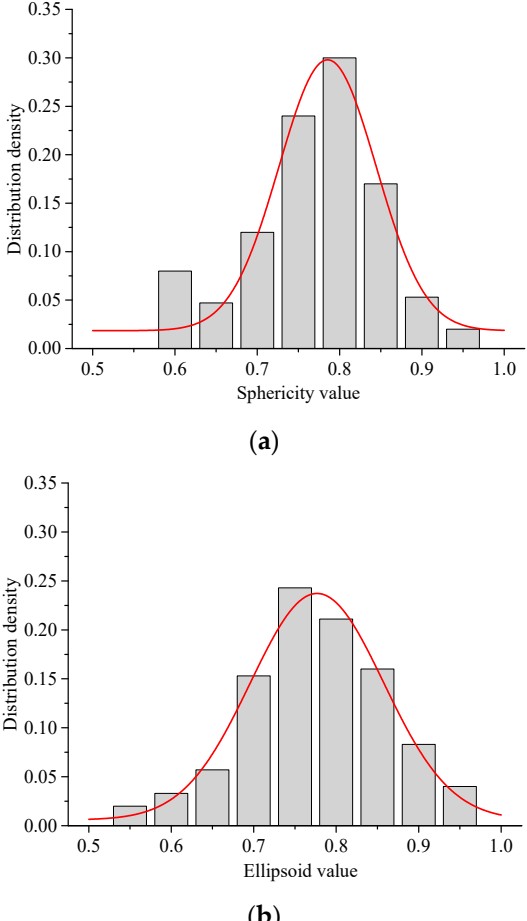

**(a)**

**(b)**

**Figure 11.** Aggregate morphology distribution of process three. (**a**) Sphericity. (**b**) Ellipticity.

From Figures 9–11 and Table 4, it can be seen that the sphericity and ellipsoid values of the aggregate shape processed by the quarry meet the Gaussian distribution, and the correlation coefficient of the fitting can reach more than 0.8.

From the point of view of the sphericity index, the average value of sphericity of process two is the largest, followed by process three and process one is the smallest, indicating that the coarse aggregate produced directly by the shaping process is closer to the cube state as a whole. The direct use of counter-breaking processing technology has a

low sphericity index, a relatively large proportion of flat particles and a large fluctuation range of particle shape among aggregate particles.

From the point of view of the ellipsoid index, the ellipsoid value of process two is the largest, followed by process three and process one is the smallest. This rule is more consistent with the sphericity index, but the ellipsoid value is slightly smaller than the sphericity value, mainly due to angularity. The ellipsoidal index of the first process is the smallest, indicating that the aggregate particles produced by counter-breaking have the best angularity, while the aggregate particles processed by the shaping process have the largest ellipticity and the weakest angularity.

In summary, the cube and angularity index reflected by the sphericity and ellipticity is a self-contradictory index. Excessive pursuit of the cube shape of coarse aggregate will cause the angularity to be easily worn during processing. The use of process three "50% counterattack crushing and 50% shaping process" is mixed with each other to achieve a balanced state of particle shape and angularity. Therefore, engineering projects can adjust the combination of aggregate processing and production technology in combination with design and use requirements, and further improve the stability of aggregate processing quality.

### 3.5. The Influence of Different Processed Aggregates on the Performance of Asphalt Mixture

There are many studies on the effect of needle-like particles on the performance of asphalt mixtures. Some scholars have found through experimental research that the volume index of the asphalt mixture is greatly affected by needle-like particles. When the needle flake particles increase, the corresponding high-temperature stability, water stability and fatigue performance of the asphalt mixture will be significantly attenuated [26,27]. However, the "Highway Engineering Aggregate Test Regulations" define needle-like particles as the ratio of aggregate length to thickness greater than 3 times. In fact, in the actual production process of the quarry, a considerable proportion of the flake crushed stone has a length to thickness ratio of 2 to 3 times. Although it can be judged as qualified, it is rich in needle-like flake aggregates or excessively shaped aggregates. The influence of the material on the performance of the asphalt mixture needs to be studied.

This article extract the aggregates produced by the previous three processing techniques and conducts the needle flake test and the mixture test. In order to keep the proportion of needle-like flakes of different process aggregates at the same level, manual removal is used to screen the excessive needle-like particles. Finally, the content of the needle-like particles of the three aggregates is in the range of 6.0% to 6.5%. The main performance indicators of the coarse material are shown in Table 5. Limestone machine-made sand is used for fine aggregate, and ordinary SBS modified asphalt (PG76-22) is used for asphalt.

**Table 5.** Coarse aggregate performance index.

| Inspection Index | Unit | Results |
|---|---|---|
| Stone crushing value | % | 6.2 |
| Los Angeles abrasion loss | % | 9.7 |
| Apparent relative density | — | 2.946 |
| Water absorption | % | 0.42 |
| Robustness | % | 1.8 |
| Water washing method < 0.075 mm particle content | % | 0.6 |
| Soft stone content | % | 0.9 |
| Adhesion to ordinary asphalt | class | 4 |

With reference to the application experience of the middle surface layer of the asphalt pavement in Guangdong Province, the framework-dense GAC-20C asphalt mixture grada-

tion is selected, and the mixing ratio parameters are shown in Table 6. The whetstone ratio is 4.4%.

**Table 6.** Gradation design of asphalt mixture.

| Size (mm) | 26.5 | 19 | 16 | 13.2 | 9.5 | 4.75 | 2.36 | 1.18 | 0.6 | 0.3 | 0.15 | 0.075 |
|---|---|---|---|---|---|---|---|---|---|---|---|---|
| Passing rate (%) | 100 | 97 | 83.6 | 70.4 | 55.7 | 33.3 | 25.1 | 17.8 | 13.7 | 10.1 | 7.3 | 5.5 |

For the asphalt pavement used in Guangdong, with a high temperature and rain in summer, the high temperature resistance of the asphalt mixture is particularly important. Three kinds of processes are used to process coarse aggregates; the rut plate specimens are formed according to the mineral gradation in Table 6, and the rutting test at 60 °C and 70 °C is carried out in accordance with the asphalt mixture test procedure. The rutting plate test piece of 300 × 300 × 50 mm is formed, and the flat rubber solid tire (outer diameter 200 mm, wheel width 50 mm) is used for round-trip rolling (the speed is 42 times/min). The contact pressure between the test wheel and the test piece is 0.7 MPa. The test temperature is 60 °C and 70 °C. An automatic rut tester (hyce-5, Beijing Aerospace Measurement & Control Technology Co., Ltd., Beijing, China) is used to measure the number of wheel rolling corresponding to 1 mm deformation of the asphalt mixture specimen, so as to characterize the high-temperature rutting resistance of the asphalt pavement. The test results are shown in Figure 12. It can be seen that the rutting test indexes of asphalt mixtures formed by coarse aggregates of different processing technologies are different. Under the condition of 60 °C, the dynamic stability index of process 3 is slightly greater than that of process 1, and process 2 is the smallest, because the dynamic stability has exceeded 6000 times/mm. At this time, the instrument and equipment display a large numerical error. Therefore, the 70 °C high temperature test is supplemented. At 70 °C, the dynamic stability value of the asphalt mixture is between 4000~5500 times/mm. Among them, the dynamic stability of the mixture of process one is the largest, followed by process three, and process two is the smallest. Therefore, it can be considered that when the aggregate needle flake indicators are qualified and the level difference is not large, the more angular the coarse aggregate, the more stable the mineral aggregate embedding, and the better the high-temperature stability of the asphalt mixture. In addition, a phenomenon is also found that the greater the angularity of the coarse aggregate, the higher the coefficient of variation of the parallel test, that is, the performance indicators of the asphalt mixture are prone to fluctuate, which is a test for the stability of construction quality.

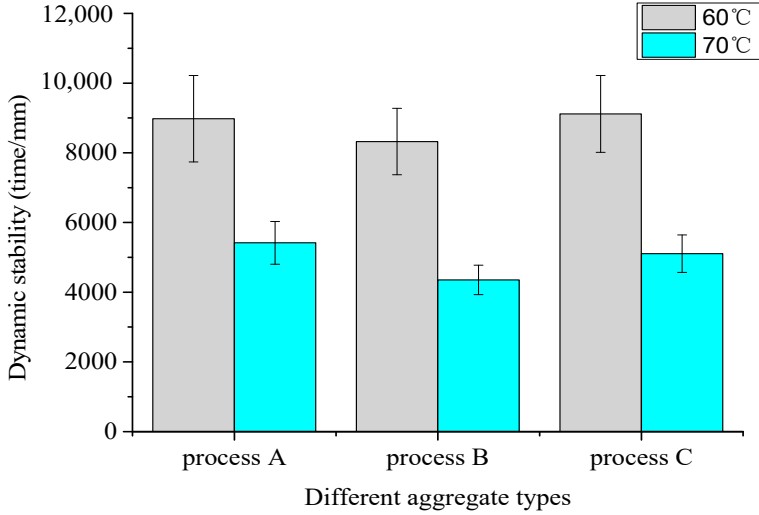

**Figure 12.** High-temperature stability of asphalt mixtures with different aggregates.

Asphalt pavement will have strength attenuation under the repeated actions of vehicles, which will cause cracking and damage. The fatigue test is generally based on the 50% initial value of the remaining stiffness modulus of the mixture as the basis for fatigue failure. The number of loading times often needs to reach dozens or even millions of times. The test volume is large, the cycle is long and the cost is high. An impact toughness test method based on the principles of fracture mechanics and the energy method is proposed [28], and the correlation between impact toughness and fatigue life is constructed. The basic test operation of impact toughness is as follows: ① Compact and shape with a wheel roller forming machine to prepare a plate-shaped specimen of 300 mm × 300 mm × 50 mm. ② Use a high-precision double-sided saw (SRC-600, SR Consulting Ltd., Klaukkala, Finland) to cut the molded solidified specimen into a length of 250 mm ± 2 mm and a width of 250 mm ± 2 mm. The prism trabecula with a height of 30 mm ± 0.5 mm and a height of 35 mm ± 0.5 mm has a span of 200 mm ± 0.5 mm. ③ Use the MTS testing machine (MTS-810, MTS Systems Corporation, Minneapolis, MN, USA) for loading at a loading rate of 500 mm/min. ④ Put the cut specimens into an environmental incubator for heat preservation and curing at 15 °C for more than 4 h and, finally, conduct an impact toughness test, as shown in Figure 13.

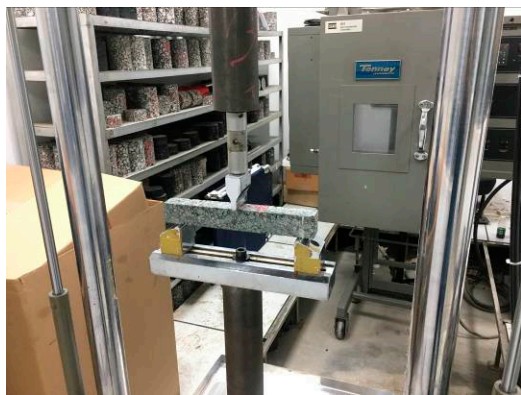

**Figure 13.** Impact toughness test.

Impact toughness mainly refers to the ability of materials to absorb deformation work and fracture work under impact load. It is an important index to evaluate the toughness of materials. When the material bears the external load, the material itself will produce a certain stress and lead to the corresponding strain. After the material produces fatigue crack under repeated load, it will produce a certain stress–strain field at the crack. Therefore, when the material breaks, it is accompanied by a loss of energy. The energy value can be calculated by the area surrounded by the load displacement diagram, that is, the impact toughness value (unit: n·mm). According to the research results of the research group on the bridge deck pavement of Hong Kong Zhuhai Macao Bridge, the correlation coefficient between the impact toughness index and fatigue performance index of asphalt mixture is more than 0.9 [29]. The impact toughness can be used to evaluate the fatigue performance of asphalt concrete. It can be seen from Figure 14 that the impact toughness value of process one is the largest, process three is the smallest and process two is the smallest. Among them, the impact toughness value of the asphalt mixture processed by process one is higher than that of the process two mixture by about 23%, indicating that the angularity of coarse aggregate is beneficial to improve the fatigue life of asphalt pavement, which in the final analysis is also related to the effect of embedding between coarse aggregates. The fatigue performance of the semi-reshaping and semi-repulsive coarse aggregate of process three is slightly lower than that of process one by about 5%, which is relatively close. In actual engineering applications, it is recommended to further combine the screening efficiency of the hot material sieve of the mixing plant and the stability of the hot material grading to further select a suitable aggregate processing technology.

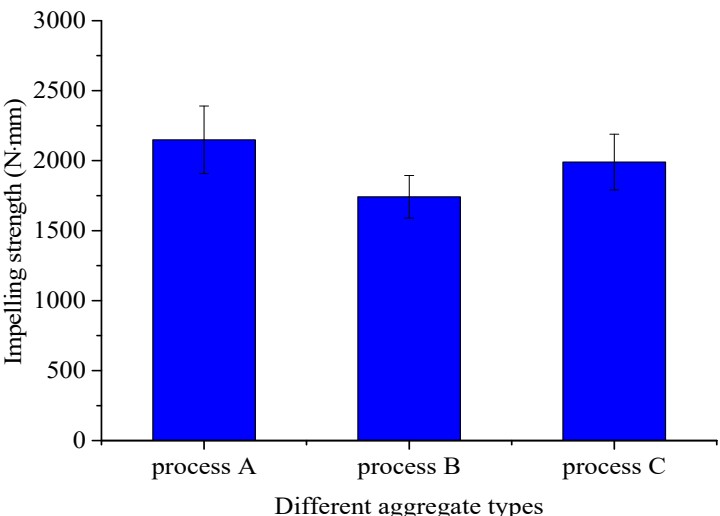

**Figure 14.** Fatigue test results of asphalt mixtures with different aggregates.

## 4. Conclusions

(1) The X, Y and Z projection profile indicators of the aggregate are analyzed. The perimeter and the projection area of the aggregate profile under different viewing angles are quite different. The closer the aggregate is to the cube, the smaller the variation in the contour index of each viewing angle; when the flatness of the aggregate is about 0.7 or more, the coefficient of variation of the two-dimensional contour index can be controlled within 5%.

(2) As the sphericity value increases, the aggregate is closer to the cubic state; the ellipsoid index calculated by the three-dimensional circumscribed ellipsoid can better characterize the angularity of the aggregate. The larger the ellipsoid value, the worse the angularity of the aggregate. Using two indicators of sphericity and ellipticity, the three-dimensional shape of the aggregate can be fully characterized.

(3) Comparing the aggregate forms processed and produced by the three quarries, the sphericity of the aggregate processed by counter-breaking is low and the angularity is better. The sphericity of the aggregate processed by the shaping process is the best but the angularity is low; it can be combined with actual needs to choose the form of combination and blending of different processing techniques to obtain aggregate finished products with a more balanced grain shape and angularity.

(4) When the aggregate needle flake indicators are all qualified and the level difference is not large, the more angular the coarse aggregate, the more stable the aggregate embedding, and the better the high-temperature stability of the asphalt mixture.

(5) The coarse aggregate produced by the impact breaking process has richer angular characteristics, and the fatigue performance of the corresponding asphalt mixture can be improved by about 23%. In practical engineering applications, it is recommended to further combine the screening efficiency of the hot material sieve of the mixing plant with the stability of the hot material grading to further select a suitable aggregate processing technology.

**Author Contributions:** Conceptualization, W.L. and D.W.; methodology, B.C.; software, B.C.; validation, K.H. and W.S.; formal analysis, C.X.; investigation, C.X.; resources, X.Z.; data curation, X.Z.; writing—original draft preparation, W.L. and W.S.; writing—review and editing, W.L.; project administration, W.L.; funding acquisition, W.L. All authors have read and agreed to the published version of the manuscript.

**Funding:** This study is funded by the "China Postdoctoral Science Foundation" (2020M672639), the China Natural Science Foundation of Guangdong Province-PhD start (No. 2018A030310684) and the Dongguan Social Science and Technology Development (General) Project (No. 2019507140574). All the help and support are greatly appreciated.

**Institutional Review Board Statement:** Not applicable.

**Informed Consent Statement:** Informed consent was obtained from all subjects involved in the study.

**Data Availability Statement:** The data presented in this study are available on request from the corresponding author.

**Conflicts of Interest:** The authors declare no conflict of interest.

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
