# Peer review of "Research on Three-Dimensional Morphological Characteristics Evaluation Method and Processing Quality of Coarse Aggregate"

_buildings, doi:10.3390/buildings12030293_

Round 1
Reviewer 1 Report
The article may address an interesting topic and provide contributions to issues not yet fully resolved concerning the characterization of aggregates for bituminous mixtures.
However, before the paper meets the conditions to be published, it is required that the authors can introduce some improvements in the text.
Regarding the tests carried out on bituminous mixtures, the authors must indicate the test conditions and applicable standards, as the procedures used are not typical for all readers.
Authors should also expressly explain how the indicators determined with 3D analysis based on the geometry of the aggregates distinguish aggregates better than the usual indices used worldwide (e.g. flakiness index; shape index, according to the European standards) for maximization properties of bituminous mixtures.
In a different perspective, I recommend the authors to show in the paper that some aggregate particles could not be used based on 3D requirements but could be used based on 2D requirements of the most common specifications. Also, show the final effect of this change on bituminous mixtures’ properties.
Author Response
Regarding the tests carried out on bituminous mixtures, the authors must indicate the test conditions and applicable standards, as the procedures used are not typical for all readers.
Answer:Thank you. According to the opinions, the relevant information of rutting test and impact toughness test of asphalt mixture is supplemented. Due to the limitation of test conditions, this study mainly adopts Chinese standards to carry out rutting test, and the test conditions are close to the British TRRL test standard. For the fatigue performance test of asphalt concrete, based on the research on the bridge deck pavement of Hong Kong Zhuhai Macao Bridge, the research group carried out the impact toughness test based on the principles of fracture mechanics and energy method, so as to evaluate the fatigue performance of asphalt pavement, which can greatly save the test time and cost. (see manuscript modification for details).
Authors should also expressly explain how the indicators determined with 3D analysis based on the geometry of the aggregates distinguish aggregates better than the usual indices used worldwide (e.g. flakiness index; shape index, according to the European standards) for maximization properties of bituminous mixtures
Answer:Thank you. Similar to China's t0311 method (using vernier caliper to measure particles with the ratio of maximum length to minimum thickness of coarse aggregate greater than 3 times), other standard methods commonly used in the world, such as British BS EN 933-4 method, mainly use a particle slide gauge to measure the shape index of aggregate, Expressed by the ratio of the mass of all particles greater than 3:1 to the total mass of each grade of aggregate. The flakiness index of aggregate is measured by British BS EN 933-3 method. Each grade of aggregate is screened by reinforcement screen. The particles passing through bar sieves are expressed by the ratio of sheet mass to total mass. These methods can characterize the needle and flake particle content of aggregate and are widely used in engineering. But there are also some limitations: first, the whole process must be completed manually, which inevitably brings subjective errors; In addition, due to the complexity of the three-dimensional shape of aggregate, it is difficult to find the accurate position of the length, width and thickness of aggregate. In contrast, the three-dimensional analysis method of aggregate can more comprehensively reflect the spatial shape of aggregate. Through the operation of the program, the maximum length, width and thickness of aggregate can also be accurately determined, and then the three-dimensional shape can be evaluated by indicators from multiple angles. (this paragraph is supplemented in the article)
In a different perspective, I recommend the authors to show in the paper that some aggregate particles could not be used based on 3D requirements but could be used based on 2D requirements of the most common specifications. Also, show the final effect of this change on bituminous mixtures’ properties.
Answer:Thank you. According to the 95% assurance rate standard in engineering application, when the flattening rate of aggregate is more than 0.7, the variation coefficient of two-dimensional contour index can be controlled within 5%. For needle flake aggregate particles, the variation coefficient of the two-dimensional contour index of the aggregate is 30% ~ 50%. At this time, the variation of the projected two-dimensional contour index is too large to fully represent the morphological characteristics of the aggregate. It is suggested that for aggregates with a flattening rate of more than 0.7, conventional two-dimensional indicators can be directly used for characterization and application. In addition, when there are too many flat particles in aggregate, it is easy to cause quality problems such as production gradation variation and construction segregation of asphalt mixture. (add at the end of 256 lines: therefore, from the evaluation method of coarse aggregate, when there are many flat particles in the aggregate, it is difficult to effectively evaluate the shape of the aggregate by using the two-dimensional contour index. It is suggested that for the aggregate with a flat rate of more than 0.7, the conventional two-dimensional index can be used for characterization and engineering application.)

Reviewer 2 Report
The manuscript “Research on Three-dimensional Morphological Characteristics Evaluation Method and Processing Quality of Coarse Aggregate” studied two-dimensional and three-dimensional morphological indicators of the aggregate from typical quarries in Guangdong Province, as well as the influence of different processing techniques on the morphology of aggregates are analyzed. I have found the methodological approach correct and the conclusions well explained. English is not bad and generally is easy to follow, but there are some evident grammar mistakes, which, in most cases, do not preclude the comprehension; hence the necessity of a revision, or not, would depend on the exigency of the journal in this aspect. To conclude, I suggest this manuscript to be published in the journal “Buildings” after the below major revisions:
- The introduction needs to be more emphasized on the research work with a detailed explanation of the whole process considering past, present, and future scope. Some references should be added in the introduction for the influence of the morphological characteristics of aggregates on their physic-mechanical properties.
- References to be cited in the introduction field:
- The Influence of the Mineralogical Composition of Ultramafic Rocks on Their Engineering Performance: A Case Study from the Veria-Naousa and Gerania Ophiolite Complexes (Greece), Geosciences, 2018.
- The Impact of Secondary Phyllosilicate Minerals on the Engineering Properties of Various Igneous Aggregates from Greece, Minerals, 2018.
- Please provide the manufacturers for all the equipment used.
- Figures 2 and 3 are really low quality.
- Is the mineralogical composition of the studied aggregates related to their morphological characteristics? A petrographic description or XRD analyses of the studied coarse aggregates could be added.
- In some parts of the manuscript, authors discussed the results in light of the literature prior to presenting their results, which is distracting.
- The conclusions of the study should be rewritten. This section is too big.
Author Response
The introduction needs to be more emphasized on the research work with a detailed explanation of the whole process considering past, present, and future scope. Some references should be added in the introduction for the influence of the morphological characteristics of aggregates on their physic-mechanical properties.
References to be cited in the introduction field:
The Influence of the Mineralogical Composition of Ultramafic Rocks on Their Engineering Performance: A Case Study from the Veria-Naousa and Gerania Ophiolite Complexes (Greece), Geosciences, 2018.
The Impact of Secondary Phyllosilicate Minerals on the Engineering Properties of Various Igneous Aggregates from Greece, Minerals, 2018.
Response:Thank you. The introduction is improved. Relevant literature is cited and summarized.
Please provide the manufacturers for all the equipment used.
Response:Thank you. Relevant equipment information has been supplemented according to the opinions.
Figures 2 and 3 are really low quality.
Response:Thank you. The HD pictures have been replaced according to the opinions.
Is the mineralogical composition of the studied aggregates related to their morphological characteristics? A petrographic description or XRD analyses of the studied coarse aggregates could be added.
Response:Thank you. For natural rocks, the hardness change of minerals, the content of soft minerals in rocks, the distribution of rock cracks and other factors have an impact on the morphology of crushed stones. Many scholars have used XRD and other technologies to carry out aggregate crystal phases of aggregate and petrological description, which is a complex and large amount of experiments. As there are few aggregate processing plants for asphalt pavement in Guangdong Province, the material sources are mainly limestone and diabase. The crushing equipment adopted for different rocks is also different. This study mainly focuses on the impact of different processing technologies on aggregate morphology in typical stone processing yards in Guangdong Province, and obtains some valuable conclusions for engineering application. Next, we will adopt the suggestions of the reviewers, investigate the mineral characteristics of different rocks, explore the impact of petrological characteristics on aggregate processing morphology, and reveal the crushing mechanism of aggregate from the micro level. (supplementary research prospect after conclusion: This study mainly studies the impact of different processing technologies on aggregate morphology in typical stone processing yards in Guangdong Province, and obtains some conclusions on engineering application. Next, we will investigate the mineral characteristics of different rocks and explore the impact of petrological characteristics on aggregate processing morphology, so as to guide the rational processing of different rocks Process design.)
In some parts of the manuscript, authors discussed the results in light of the literature prior to presenting their results, which is distracting.
Response: Thank you. Relevant issues have been modified.
The conclusions of the study should be rewritten. This section is too big.
Response:Thank you. The conclusion is re sorted according to the opinions.
Round 2
Reviewer 2 Report
The authors carefully followed the comments and suggestions, made appropriate corrections and the manuscript in the present form was sufficiently improved with respect to the previous version. I recommend to accept this manuscript for publication.
Author Response
Thank you.
